# Unraveling Bicuspid Aortic Valve Enigmas by Multimodality Imaging: Clinical Implications

**DOI:** 10.3390/jcm11020456

**Published:** 2022-01-17

**Authors:** Arturo Evangelista Masip, Laura Galian-Gay, Andrea Guala, Angela Lopez-Sainz, Gisela Teixido-Turà, Aroa Ruiz Muñoz, Filipa Valente, Laura Gutierrez, Ruben Fernandez-Galera, Guillem Casas, Alejandro Panaro, Alba Marigliano, Marina Huguet, Teresa González-Alujas, Jose Rodriguez-Palomares

**Affiliations:** 1Departament de Cardiologia, Hospital Vall d’Hebron.CIBERCV, Universitat Autonoma de Barcelona, 08035 Barcelona, Spain; lauragaliangay@gmail.com (L.G.-G.); andrea.guala86@gmail.com (A.G.); kelals@hotmail.con (A.L.-S.); gisela.tt@gmail.com (G.T.-T.); aroa.ruiz.munoz@gmail.com (A.R.M.); filipaxaviervalente@gmail.com (F.V.); lauraguga@gmail.com (L.G.); ruben.fernandez@vhebron.net (R.F.-G.); gcasasmasnou@gmail.com (G.C.); teresagonzalu@gmail.com (T.G.-A.); jfrodriguezpalomares@gmail.com (J.R.-P.); 2Vall d’Hebron Institut de Recerca (VHIR), 08035 Barcelona, Spain; 3Teknon Heart Institute-Quiron Salud, 08022 Barcelona, Spain; alejandro.panaro@quironsalud.es (A.P.); albamarigliano@gmail.com (A.M.); marina.huguet@quironsalud.es (M.H.)

**Keywords:** bicuspid aortic valve, echocardiography, computed tomography, magnetic resonance imaging, aortic aneurysm

## Abstract

Multimodality imaging is the basis of the diagnosis, follow-up, and surgical management of bicuspid aortic valve (BAV) patients. Transthoracic echocardiography (TTE) is used in our clinical routine practice as a first line imaging for BAV diagnosis, valvular phenotyping and function, measurement of thoracic aorta, exclusion of other aortic malformations, and for the assessment of complications such are infective endocarditis and aortic. Nevertheless, TTE is less useful if we want to assess accurately other aortic segments such as mid-distal ascending aorta, where computed tomography (CT) and magnetic resonance (CMR) could improve the precision of aorta size measurement by multiplanar reconstructions. A major advantage of CT is its superior spatial resolution, which affords a better definition of valve morphology and calcification, accuracy, and reproducibility of ascending aorta size, and allows for coronary artery assessment. Moreover, CMR offers the opportunity of being able to evaluate aortic functional properties and blood flow patterns. In this setting, new developed sequences such as 4D-flow may provide new parameters to predict events during follow up. The integration of all multimodality information facilitates a comprehensive evaluation of morphologic and dynamic features, stratification of the risk, and therapy guidance of this cohort of patients.

## 1. Introduction

Bicuspid aortic valve (BAV) is the most common congenital heart disease, with a reported prevalence of 0.8–1.5% [1,2] and a male predominance of nearly 3:1. It is considered to be a valvulo-aortopathy characterized by a large individual heterogeneity, both at the valvular and aorta level, but also in the possible associated disorders, complications, and prognoseis [3,4]. The most frequent complication of the BAV condition in adults is aortic valvular dysfunction, more frequently in the form of severe aortic stenosis (AS), needing surgical aortic repair or AV replacement (AVR) (population-based 25-year risk of AVR is up to 50%) [1]. The second most frequent complication, especially in those above 60 years-old, is ascending aorta dilation. However, the important heterogeneity in terms of the pattern of aortic dilation described in the complete BAV cohort is possible heterogeneity in the molecular, rheologic, and clinical features [5,6,7,8,9].

Early and accurate diagnosis of BAV and its complications by imaging techniques is essential to improve the management and outcome of BAV patients. Transthoracic echocardiography (TTE) is used in clinical practice for first line imaging for BAV diagnosis, valvular phenotyping and function, measurement of thoracic aorta, exclusion of other aortic malformations, and assessment of complications such are infective endocarditis and aortic dissection The aim of this review was to update the role of multimodality imaging techniques in BAV assessment, including recent advances and pending gaps.

The main objectives of imaging techniques in the evaluation of this disease are: (1) BAV diagnosis; (2) phenotype description; (3) valvular function assessment; (4) ascending aorta dilation measurement; (5) exclusion of aortic coarctation; and (6), in some specific contexts, ruling out the presence of severe complications such as endocarditis and aortic dissection. In fact, multimodality imaging is the cornerstone not only in diagnosis, but it is also a relevant tool during follow-up of this congenital disease in order to evaluate the necessity of surgery intervention, and to assess post-surgical surveillance.

Several classifications have been proposed for the BAV condition [9,10,11,12,13,14,15,16,17], most of which include multiple numbers and letters for the BAV and aorta phenotypes [9]. Recently, Michelena et al. published an International Consensus Statement on Nomenclature and Classification of BAV and its aortopathy [18]. This consensus recognized three types of bicuspid valve: the fused type (right-left cusp fusion, right-non-coronary cusp fusion and left-non-coronary cusp fusion phenotypes), the two-sinus type (latero−lateral and antero−posterior phenotypes), and the partial-fusion (forme fruste) type. The authors point to the presence of raphe and the symmetry of the fused type phenotypes as the most important aspects to describe. The International Consensus also recognizes three types of bicuspid valve-associated aortopathy—the ascending phenotype, root phenotype, and extended phenotype.

## 2. Diagnosis and BAV Phenotype

Transthoracic echocardiography (TTE) is the first imaging tool to diagnose the presence of BAV, given its accuracy in 80–90% of cases [1]. Nevertheless, there are some limitations we should take into account, like the extensively-calcified valve, poor visualization of the valve, or the distal ascending aorta. Other imaging techniques such as computed tomography (CT) or cardiac magnetic resonance (CMR) may solve these drawbacks. Transesophageal echocardiography (TEE) is also useful for detecting BAV; however, we must remember it is semi-invasive and it also has some limitations in order to assess the upper part of the ascending aorta and the proximal arch. 

The BAV phenotypic expression represents an anatomic continuum of increasing non-fused cusp commissural angles and increasing similarity of cusp size and shape. This spectrum goes from the partial-fusion BAV (very near to a tricuspid aortic valve) to other phenotypes such us those asymmetric fused, symmetric fused phenotypes with and without a raphe, and finally to the two-sinus BAV, which is considered the most severe defect and is anatomically close to perfect “bicuspidity” [19]. The best view of the phenotype BAV is the TTE parasternal short-axis or its equivalent in TEE, CT, or CMR (Figure 1).

The most common type of BAV is the fused one (90–95% of the cases) [19]. This is described as two of the three cusps appearing fused within three aortic sinuses, resulting in two functional cusps commonly different in shape and size. A congenital fibrous ridge is often described between the fused cusps, and it is what we know as “raphe”. The most frequent BAV phenotype is the one with right−left (R−L) fusion, considered to be present in 75% of the cohort. This is followed by the right−non-coronary (R−N) fusion (20–25%), and finally the left−non (L−N) fusion (<3%). The two-sinus BAV type is uncommon and accounts for approximately 5–7% of cases. In contrast to the fused type, the two-sinus BAV appearance suggests that two roughly equal size-shape cusps, each cusp occupying 180 degrees of the annular circumference, are “formed” within only two aortic sinuses, resulting in a two-sinus/two-cusp valve [20] with a latero−lateral (side-to-side) or antero−posterior (front-and-back) position. The presence of an incomplete fusion of two leaflets (mini-raphe) cannot be easily visualized by TTE [21] (Figure 2). In these cases, we should use other techniques such as TEE and CT. This mini-raphe has been considered to be involved in aorta dilation [22].

In order to solve the limitations of the TTE, CMR and CT are very useful in the morphologic valve evaluation when it is heavily calcified, and can readily assess aorta diameters [23]. CT can quantify the valve calcification, which can help clinicians to evaluate valve stenosis severity. The multiplanar reconstructions permit CMR and CT to precisely assess the aortic diameters [24,25]. Although CT offers a higher spatial resolution, CMR provides information of the valvular dysfunction and left ventricular function. Cine images (SSFP sequences) can be used for measuring the luminogram of the aorta, particularly in the aortic root, owing the valvular plane movement. Aorta diameters should be measured in double oblique technique) [26] (Figure 3). Screening for coarctation should be performed in BAV patients, however CMR on the first echo is the preferred technique in younger patients in order to avoid radiation.

## 3. Familial Screening

BAV is diagnosed in 4.6–11% of first-degree relatives (FDR) [27,28,29,30]. Moreover, the frequency of aorta dilation in FDR is nearly 10% in those with tricuspid aortic valves. In a recent publication by our group, we reported the presence of a mini-raphe by CT in 41% of cases with aorta dilation in FDR and an apparent absence of BAV by TTE [31], and further studies demonstrate that, compared with BAV and tricuspid valve patients, the presence of a mini-raphe is associated with ascending aorta flow pattern alterations with increased flow eccentricity and increased vorticity [22]. These data are in favor of the screening in the FDR of BAV patients.

## 4. Valvular Dysfunction

In adult patients with BAV, the most frequent complication is valvular dysfunction, commonly known as aortic stenosis (AS), which necessitates surgical aortic valve replacement (AVR) or repair [3,32,33]. TTE is the technique of choice in the diagnosis and quantification of aortic valve dysfunction. The subtype with a lower prevalence of valvular dysfunction is pure BAV without raphe. This subgroup showed both a lower prevalence of AS, as well as of aortic regurgitation (AR), in a large international registry that included 2118 patients [34]. If we differentiate in terms of age, AR has been described to be more frequent in young individuals, and AS more frequent in the older ones [35].

### 4.1. Valve Calcification

BAV calcification has been associated with inflammatory processes mediated by hemodynamic, molecular, and genetic factors [36]. Calcification is usually localized in the raphe, and it is considered to be a risk factor for developing valve degeneration and aortic stenosis (AS). Stenotic calcification of a BAV could appear more than ten years earlier than in those with the tricuspid aortic valve. It is related to conventional cardiovascular risk factors. An echocardiographic semiquantitative score for aortic valve calcification has been validated [37]. CT calcium scoring is useful for valve stenosis quantification in doubtful cases. In a large crossectional study of 852 BAV [35] patients, we found age, arterial hypertension, dyslipidemia, smoking, and the BAV-RN morphotype to be associated with aortic valve calcification. However, CT score was not associated with valve phenotype [38].

### 4.2. Aortic Stenosis

Adults diagnosed with BAV have a clear increased risk of developing AS, usually secondary to leaflet calcification. Previous studies have showed that over 15 years of follow up, 12.3% of the cohort required aortic valve replacement for severe AS [32]. In our daily practice, TTE is the most used technique for evaluating the presence of this complication, and for guiding appropriate management. AS quantification is based on the same parameters of tricuspid valves (Figure 4). We must remember that the aortic valve area may be significantly underestimated by TTE owing to underestimation of aortic annulus measurement. However, the use of CTA may correct these underestimations [1].

Although in some studies it has been suggested that AS progresses faster in BAV than in the tricuspid aortic valve [39], a recent large study by Michelena et al. [40] showed that BAV and tricuspid aortic valve stenosis have similar progression rates, with evidence of accelerated progression (non-linearity) in those attaining severe AS. Determinants of rapid progression in BAV-AS, again, are modifiable cardiovascular risk factors, particularly for patients with BAV who are <60 years of age.

### 4.3. Aortic Regurgitation

Valvular dysfunction as AR is more frequent in younger patients, but also in males [41]. Although its prevalence ranges between 47% and 64%, moderate−severe grade appears in less than 30% of individuals [9,32,33,34,35,42]. Annular dilation and cusp prolapse or retraction is the most common underlying cause of chronic regurgitation, acting either alone or in combination [43]. The evaluation of AR severity in BAV by TTE is challenging, given that eccentric jets are common. In some cases, CMR may be superior to TTE for quantifying AR (Figure 5) [44,45]. The CMR phase contrast is accurate and reproducible in AR assessment with the estimation of the regurgitant volume and regurgitant fraction. We labelled a regurgitant fraction of more than 30% as severe [46].

## 5. Aorta Involvement

Ascending aorta dilation has been frequently described in patients with BAV. Previous studies have shown it to be present in 56% of patients under the age of 30, but up to 88% in those over 30 [32,33,42]. In daily practice, TTE is the most used technique to assess the dilatation of the proximal ascending aorta. Although the aortic root and proximal ascending aorta are well visualized in the left parasternal long-axis view, the mid ascending aorta should be visualized by moving the transducer position to an upper intercostal space. Ascending aorta diameter by TTE using leading–leading edges yield comparable values to inner–inner edges by CT and CMR [47]. However, failure to obtain the largest diameter by TTE can lead to an underestimation of the aortic root diameter. This drawback may be important in the presence of root asymmetry (≥5 mm between cusp-to-cusp diameters), which is described in more than 50% of BAV patients without raphe and in >40% of the BAV-RN morphotype [48]. Therefore, BAV-RN and cases without raphe may particularly benefit from a CMR/CT study to ensure the true largest aortic root diameter (Figure 3B).

Dilation of the mid ascending segment with normal or mild aortic root dilation is observed in 70–80% of patients, and predominant dilation of the aortic root is observed in 20–30% [48]. BAV-RN morphotype spares aortic root dilation and shows distal ascending of the aorta, while BAV-LR more commonly dilates the aortic root and spares the arch [35]. AR is independently associated with aortic root dilation [49].

Table 1 shows a comparison of the imaging modalities for diagnostic features of bicuspid aortic valve. In Figure 6, the role of multimodality imaging for the diagnostic features in bicuspid valve is specified.

## 6. Imaging Predictors

Very few studies have assessed the progression of valvular dysfunction and ascending aorta dilation in BAV patients. Data regarding the association between BAV leaflet morphology and AS are conflicting. Beppu et al. [50] reported that BAV-LR was more likely associated with the rapid development of AS. Nevertheless, in a larger study, Kong et al. [34] reported that BAV-RN and BAV-LN have a higher incidence of significant AS. The population-based longitudinal study by Michelena et al. [51] failed to identify any relationship between BAV morphotypes and AS upon follow-up. Regarding the progression of AR, the only variables were males, arterial hypertension, and sigmoid prolapse reported in transverse studies [35,52].

Multiple studies [53] have demonstrated that the segment that presents the maximum growth rate is the tubular ascending aorta (0.4 to 0.6 mm per year). This rate is influenced by age, baseline aorta diameter, associated valve dysfunction (regurgitation vs. stenosis), and location of the dilation (ascending versus root) [54,55]. Accumulated evidence shows that the root phenotype might represent a more severe form of aortopathy, possibly related to a genetic underlying condition that could contribute to the pathogenesis, especially if compared with the ascending phenotype. Furthermore, this root phenotype has been described to be associated with aortic events in the postoperative period of those who had undergone simple aortic valve replacement (Figure 7).

### 6.1. Aorta Biomechanics Predictor Parameters

In order to non-invasively evaluate thoracic aorta stiffness, we can measure distensibility, but also the pulse wave velocity of the propagation of flow by CMR. 

Aorta distensibility assesses the relative change in the cross-sectional area during the cardiac cycle divided by the local pulse pressure. Aorta distensibility is most often assessed in the ascending aorta and less-often in the aortic root, owing to the difficulty generated by its motion and non-circular area [56]. Ascending aorta distensibility has been found to be lower in predominantly-dilated BAV patient groups compared to the healthy population [56,57,58,59].

Pulse wave velocity (PWV) depends mainly on wall stiffness and local diameter: the stiffer and/or smaller an artery is, the faster waves propagate in it. PWV showed the best association with ascending aorta dilation in BAV patients, beyond the clinical risk factors. The most significant advantages of PWV are that, unlike aortic distensibility, it does not rely on any geometric assumptions and it is not based on local pressure. A recent study [56] reported regional PWV by 4D-flow CMR in BAV patients. Non-dilated BAV patients had a similar PWV to healthy controls in both the ascending and descending aorta, whereas dilated BAV patients did not differ from dilated tricuspid valve patients. Furthermore, the relationship between PWV and tubular aorta dilation was biphasic: first it decreased and then it increased throughout dilation with a clear turning point at 50mm, which may highlight the end of aortic adaptation to dilation severity [56].

### 6.2. Aortic Flow Hemodynamic Parameters

4D-flowCMR is very useful in the evaluation of abnormal ascending aorta flow in BAV patients (Figure 8). Several studies have shown the flow in the aorta of BAV patients to be eccentric, not directed in the same direction of the aorta [60,61,62,63]. Such flow abnormalities, also reported in non-dilated aortas, are thought to produce changes in wall shear stress (WSS), the force per unit of area exerted tangentially by blood on the aortic wall, which is associated with elastic fiber degeneration and extracellular matrix dysregulation [64]. The 4D-flow MRI shows different local flow abnormalities associated with the different leaflet fusion patterns [60] in particular, unlike at the ascending tract, where average systolic wall shear stress with BAV-RL and BAV-RN reached similar magnitudes. Higher shear stress values were found at the more proximal tract of the aorta with BAV-RL. However, the RN morphotype spared dilation at the root level and showed higher in-plane shear stress in cross sections at the tubular distal level. Interestingly, partial aortic valve leaflet fusion, a forme fruste BAV, may alter aortic flow patterns and may lead to aorta dilation [22,65]. In a longitudinal study, our group proved that in patients with BAV without significant valvular dysfunction, circumferential wall shear stress component independently predicted progressive dilation of the ascending aorta. Notably, mapping of WSS further allowed for establishing ascending aorta regions with the fastest dilation rates [66].

## 7. Imaging Surveillance

In clinical practice, the modality used for the serial evaluation of valvular dysfunction is TTE, as well as for the measurement of maximum diameters, thus helping to determine the timing of elective surgery. Follow-up by imaging test is different depending on the baseline valvular dysfunction severity. In mild abnormalities, a control every 3–5 years may be enough, while a 6–12 month interval is more advisable in severe cases. In patients with root or tubular ascending aorta diameter >40 mm, CT or CMR should be performed to evaluate the correlation with the echo measurements. If measurements are comparable among techniques, then future interval measurements can be obtained by TTE with repeat CT or CMR every 3–5 years. If the initial measurements are discrepant (>3 mm), CT or CMR should be the technique of choice. In more advanced aorta dilation (>50 mm), imaging control should be performed at least yearly. Measurements should be taken with the same imaging modality and technique (i.e., ECG-gated) and be compared side-by-side by an experienced reader. The tubular ascending aortic growth rate was recently reported to range from 0.4 to 0.6 mm/year [67]. Although these represent “artificial-annualized” rates, it remains unlikely that patients will dilate at more than 3 mm/year. It is also important to note that a diameter change of 2 mm is within the margin of error by current imaging modalities [68]. Therefore, an interval dilation of ≥3 mm should be considered clinically significant. Recently, a semi-automatic technique to map aortic dilation rate from pairs of CTs was presented and validated, allowing for a reliable assessment of the growth rate from a much shorter follow-up duration [68].

## 8. Complications: Surgical and TAVI Indications

The most common complications in patients with BAV are aortic valve replacement, which occurs in over 50% of patients, followed by ascending aortic aneurysm formation, occurring in over 25%. In the population-study reported by Michelena et al. [51], the incidence in 20 years of follow-up of aortic valve replacement was 25 ± 4% (70% of them due aortic stenosis). In the same series, the incidence of ascending aorta replacement was 5 ± 2% and surgery for aortic coarctation was 4 ± 1%. Coarctation of the aorta is present in 7–10% of adults with BAV, whereas BAV is present in 50–60% of patients with coarctation. Concomitant coarctation is associated with a higher risk of aortic complications [25]. The least frequent, but most deadly, complications are infective endocarditis and aortic dissection.

### 8.1. Infective Endocarditis

Infective endocarditis is a severe complication in BAV patients with an incidence of 9.9 (95% CI, 4.4–22) per 10,000 patient-years, resulting in an age-adjusted relative risk of IE for those with native BAVs of 16.9 (95% CI, 7.6–37.6) compared with the general Olmsted County population (*p* < 0.0001) [4]. Prompt diagnosis should be made to improve prognosis, as in BAV, infective endocarditis incur a greater severity of complications, like a higher risk for the formation of abscesses. The echocardiogram remains the gold standard technique to rule out the presence of infective endocarditis. The sensitivity of TTE for the detection of vegetations in a BAV ranges from 50 to 90% [69]. Thus, initial negative images do not definitely dismiss the diagnosis. When clinical suspicion is high, patient should undergo a TEE, which in these cases has a sensitivity of 90–100% [69]. This technique also permits the diagnosis of secondary complications such as abscesses or paravalvular involvement.

### 8.2. Aortic Dissection

The most feared BAV complication is aortic dissection, with a reported incidence of 3.1 cases per 10,000 patient-years, which is eight times that of the general population, increasing to 0.5% in patients with aortic diameters >45 mm. Michelena et al. [70] published a series of 47 BAV patients with aortic dissection and found that the maximum ascending aortic diameter at aortic dissection was higher in patients with BAV (66 ± 15 mm vs. 56 ± 11 mm, *p* = 0.0004). Furthermore, previous aortic valve replacement was more common in BAV (23% vs. 6%, *p* = 0.02) and 23% had a history of previous repair coarctation. Although a diagnostic of aortic dissection is feasible by TTE when the intimal flap is visualized in the proximal ascending aorta, CT is the technique of choice. TEE plays a crucial role in monitoring surgical treatment in the operating room.

### 8.3. Aortic Stenosis: Surgical and TAVR Indications

Surgical treatment indications of valvular dysfunction are based on TTE information and are similar to the tricuspid aortic valve [71]. In aortic stenosis, intervention is recommended in symptomatic patients with severe, high-gradient aortic stenosis with a mean gradient ≥40 mmHg, peak velocity ≥4.0 m/s, and valve area 1.0 cm^2^ (or ≤ −0.6 cm^2^/m^2^). Similarly, in patients with severe low-flow (stroke volume index ≤35 mL/m^2^) low-gradient (<40 mmHg) aortic stenosis with reduced ejection fraction (<50%). In asymptomatic patients, surgery is also indicated with LVEF >55% if the procedural risk is low and one of the following parameters is present: mean gradient ≥ 60 mmHg or maximum velocity > 5 m/s, severe valve calcification and velocity progression ≥ 0.3 m/s/year, or markedly elevated BNP levels without other explanation.

Nowadays, TAVR is an established therapy option for patients with symptomatic severe AS. Although the first randomized trials considered BAV as an exclusion criterion, real-life registries have shown that a significant number of patients have already been treated with TAVR [72]. It has been reported that this intervention, when performed in patients suffering from BAV stenosis, is associated with higher rates of paravalvular regurgitation, early mortality rates, and a likelihood of conversion to surgery. This poor prognosis is mainly related to the anatomical characteristics of BAV: more elliptical annulus, enlarged aortic root and ascending aorta, and unequally sized leaflets along with asymmetric leaflet calcification. All of this information is well defined by CT angiography. The recent BAV stenosis TAVR Registry [73] found that outcomes of TAVR in bicuspid aortic stenosis depend on the valve morphology. Calcified raphe and excess leaflet calcification are associated with increased risk of procedural complications and midterm mortality. The design of new-generation devices, and the increasing use of multidetector CT scan to preoperatively size the aortic annulus have contributed to the near disappearance of these fearsome complications. In fact, the international registry on TAVR in BAV stenosis reported no significant paravalvular regurgitation when new generation devices are used [73], as well as lower rates of annulus rupture or conversion to surgery [74].

### 8.4. Aortic Regurgitation: Valve Repair and Replacement

In aortic regurgitation, aortic valve replacement is recommended in asymptomatic patients with end-systolic diameter by TTE > 50 mm or this diameter indexed by body surface area > 25 mm/m^2^, with an ejection fraction ≤ 50%.

Valve-preserving surgery and repair of regurgitant BAV have evolved into an increasingly used alternative to replacement [75]. TEE plays a critical role in determining the reparability of the regurgitant BAV, which is successful more frequently in BAV than in tricuspid aortic valves [76], with a low cumulative reoperation incidence of 20% at 15 years when combined with root remodeling. Typically, the main mechanism leading to BAV regurgitation is prolapse of the fused cusp (for fused BAV types) [77,78]. Assessment of BAV symmetry for the fused BAV type, defined by the angle between the commissures of the non-fused cusp, is key information in surgical BAV repair for pure aortic regurgitation. The orientation of the two true commissures affects post-repair outcomes; when the commissures are located 160° to 180° apart, leaflet stresses are lower, and 10-year freedom-from-reoperation rates can be more than 40% higher than in patients with less than a 160° commissural orientation [79].

Anatomic predictors of possible repair failures have been well identified and solutions have been developed [80]. Using current techniques, most non-calcified BAVs can be preserved or repaired. Excellent repair durability and freedom from valve-related complications can be achieved if all pathologic components of aortic valve and root including annular dilation are corrected.

Calcifications of the raphe or cusp retraction (geometric height < 20 mm) that cannot be treated by direct approximation of the cusp tissue are better treated by replacement. An unfavorable commissural orientation will increase the complexity of the repair and decrease its durability [75]. Although asymmetric or very asymmetric BAV can be repaired, they represent a higher technical challenge for the surgeon. Using current approaches for the selection of repairable valves and also for repair strategies, the isolated repair of a regurgitant BAV has been possible, with a 10-year freedom from reoperation of 90% [75].

### 8.5. Ascending Aorta Surgery

The European Society of Cardiology [71] and the American Association for Thoracic Surgery consensus [67] document recommended surgery for aortic dilation exceeding 55 mm in BAV patients similar to the general population. Replacement of the ascending aorta/root should be performed when aorta diameter is ≥50 mm in patients with risk factors (i.e., root phenotype or predominant aortic regurgitation, uncontrolled hypertension, family history of aortic dissection, coarctation, and aorta growth-rate >3 mm/year). In patients who are candidates for surgery based on valve dysfunction, it is recommended to replace the root and/or ascending aorta when the maximum diameter is >45 mm.

An indexed aortic diameter cut-off of greater than 2.75 cm/m^2^r and an aortic cross-sectional area to height ratio of greater than 10 cm/m^2^ may be used to guide earlier surgical intervention in patients with small stature; earlier intervention is occasionally justified in patients with a strong preference for early surgery. Finally, surgical repair may be performed at a lower threshold (i.e., 5.0 cm) if the patient has a low operative risk and the procedure is performed by an experienced operative team with established results.

## 9. Critical Knowledge Gaps

Despite significant advances in understanding BAV disease in the last decade, many important questions remain to be answered. Why one BAV becomes stenotic, another regurgitant, another is associated with aortic dilation, and yet another continues to be functional throughout a lifetime, is fundamentally unknown and unpredictable, and remains unresolved.

Research should be implemented in order to: (a) individualize the imaging follow-up and establish the multimodality protocol to recommend lifestyle and surgical treatment indication; (b) define the predictors of valve dysfunction, especially aortic valve calcification and aorta dilation; (c) specify whether the indication for surgery should be based on the aorta diameter in its absolute value or indexed by body surface area or by height; (d) confirm that aortic root dilation with associated regurgitation presents a greater risk of aortic dissection and therefore advance the indication for surgery when its diameter is ≥50 mm; and (e) determine which the imaging biomechanical parameter can independently predict the rapid progression of aortic dilation.

Considering the high prevalence of BAV and the marked heterogeneity of its evolution, it is important to identify patients with a rapid evolution profile compared to more stable ones. Imaging techniques should provide this information, which would facilitate better personalization of follow-up, lifestyle, possible medical treatment, and better timing in the indication of surgical treatment.

## 10. Highlights


-Multimodality imaging is the cornerstone of the diagnosis, follow-up, and surgical treatment indication in BAV patients.-Although TTE is the technique of choice to identify valve morphotype, valve dysfunction, and ascending aorta dilation, CT and CMR improve the accuracy of aorta size measurement.-A major benefit of CT is its superior spatial resolution, by which the aortic valve can be evaluated for morphology and calcification, ascending aorta size accuracy, and reproducibility measured and coronary arteries assessed.-CMR offers the unique opportunity to transition from anatomic to dynamic imaging of the ascending aorta by assessing its functional properties and blood flow patterns. 4D-flow CMR may provide new parameters to predict the rate of progressive aorta enlargement.-Therefore, a multimodality approach may offer a comprehensive assessment of morphology, risk stratification, and therapy guidance of BAV disease.


## Figures and Tables

**Figure 1 jcm-11-00456-f001:**
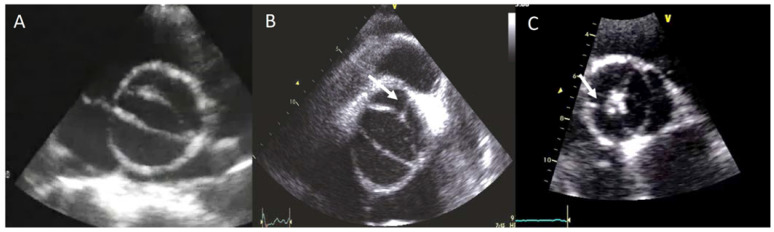
Most frequent types of BAV by TTE. (**A**) BAV without raphe type antero−posterior; (**B**) RC−LC fusion with raphe (arrow); (**C**) RC−NC fusion with raphe (arrow).

**Figure 2 jcm-11-00456-f002:**
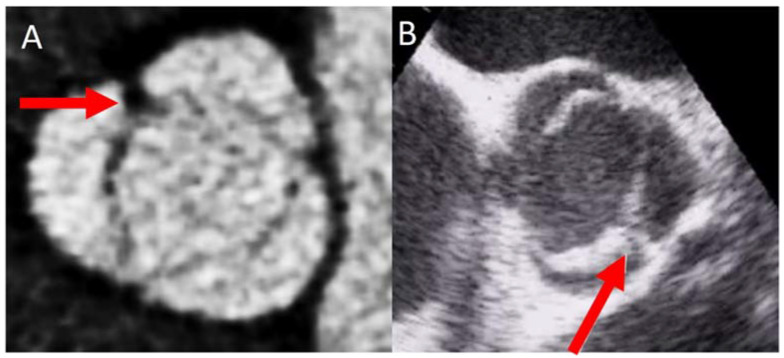
BAV with partial fusion (forma frustre). Arrows show the mini raphe by CT (**A**) and by TEE (**B**).

**Figure 3 jcm-11-00456-f003:**
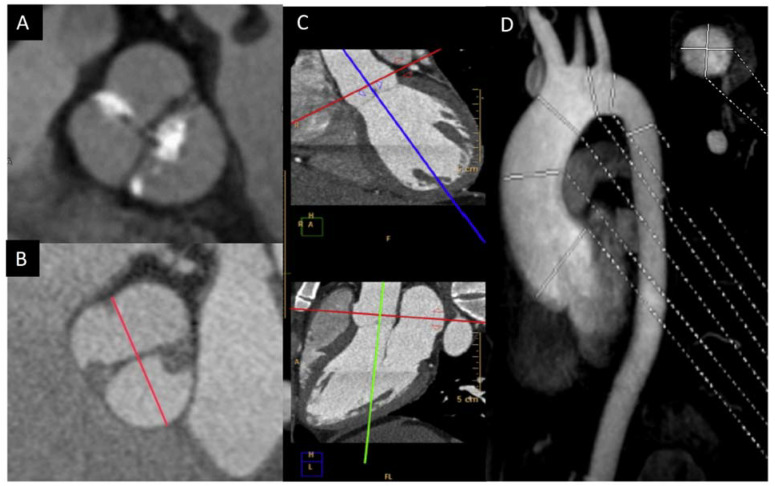
BAV by CT showing left−right fusion with raphe calcification and mild non-coronary sigmoid edge calcification (**A**); CMR showing two-sinus anteroposterior BAV (**B**); double obliquity image for measuring the maximum diameter of the aortic root by CT (**C**); thoracic aorta diameters by angio-CMR sagittal projection; the right upper part shows the aortic root section obtained with double obliquity image (**D**).

**Figure 4 jcm-11-00456-f004:**
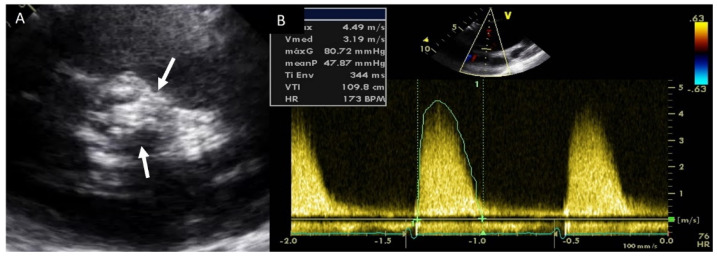
Severe aortic stenosis in BAV with severe calcification (arrows) (**A**); the mean gradient by continuous-wave Doppler is 48 mmHg (**B**).

**Figure 5 jcm-11-00456-f005:**
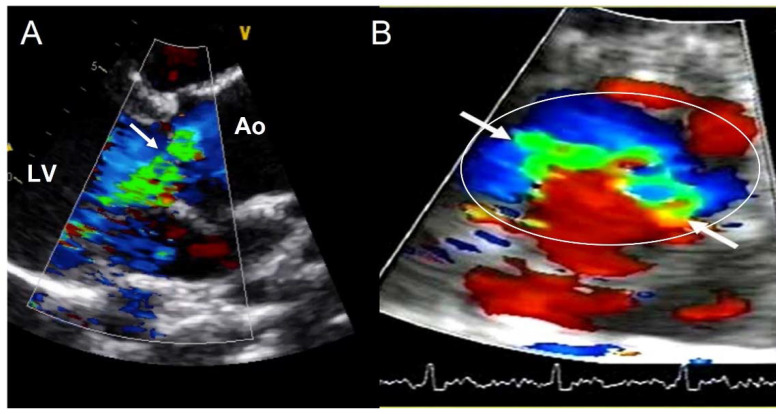
Severe aortic regurgitation in a BAV. (**A**) Eccentric jet in parasternal long axis-view (arrow). (**B**) The short-axis view shows the elliptic shape of the regurgitant orifice (arrows) in the aortic annulus (circle). Ao—aorta; LV—left ventricle.

**Figure 6 jcm-11-00456-f006:**
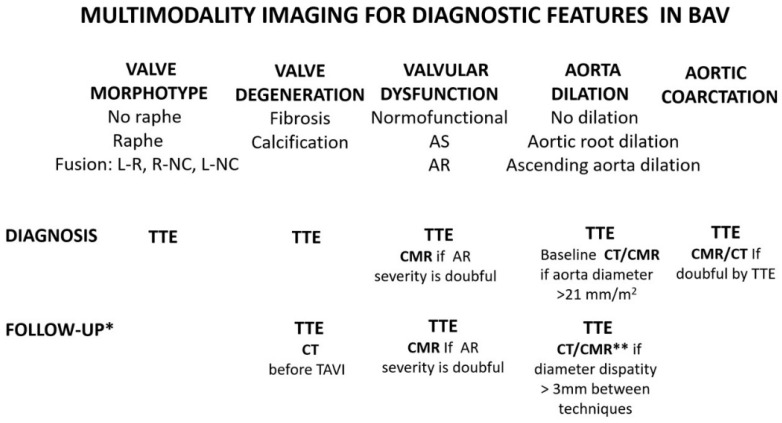
Multimodality imaging for diagnostic and follow-up of features in bicuspid aortic valve patients. * Follow-up depending on the severity of valvular dysfunction and aorta dilation. ** When disparity between TTE and CT/MRI, >3 mm follow-up should be performed annually by TTE and every 3 years by CT/MRI if the maximum diameter is <45 mm, every 2 years if it is between 45–50 mm, and yearly if >50 mm. Abbreviatures: TTE: transthoracic echocardiography, MRI: magnetic resonance imaging; CT—computed tomography; Fusion L-R—left-right sigmoids fusion; R−NC—right−non coronary sigmoids; L−NC— left-non coronary sigmoids. AS—aortic stenosis; AR—aortic regurgitation.

**Figure 7 jcm-11-00456-f007:**
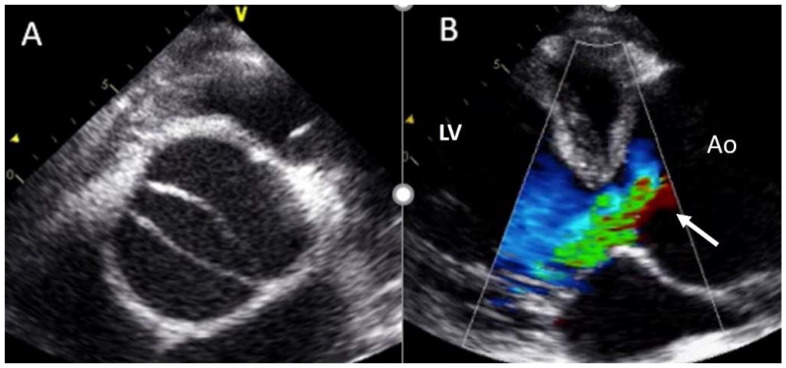
(**A**) BAV RC−LC sigmoid fusion in a patient with Marfan syndrome. (**B**) Severe aortic root dilation and severe aortic regurgitation (arrow). Ao—aortic root; LV— left ventricle.

**Figure 8 jcm-11-00456-f008:**
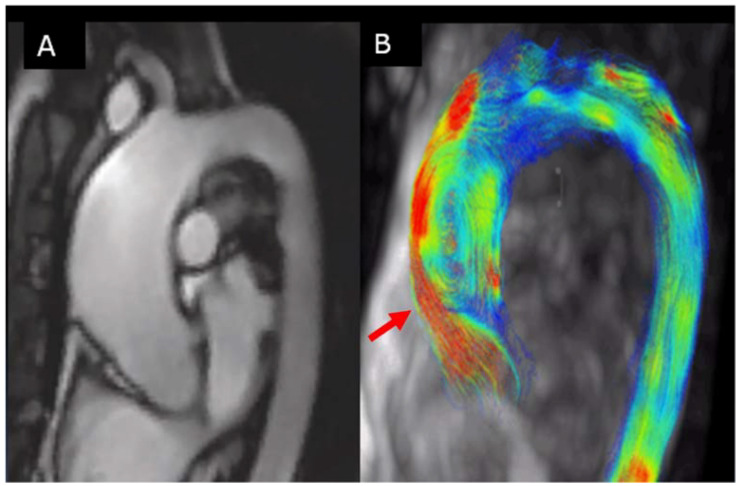
(**A**) Thoracic aorta by CMR images of BAV with RC−LC fusion; (**B**) 4D-MRI dimensional flow. Note that flow impinges on the outer curvature of the proximal ascending aorta (arrows), including the root (red arrow).

**Table 1 jcm-11-00456-t001:** Comparison of imaging modalities for diagnostic features of bicuspid aortic valve. TTE—transthoracic echocardiography; CT—computed tomography.

	TTE	CT	MRI
VALVE MORPHOTYPE	+++	++	+++
RAPHE	+++	+++	++
VALVE DEGENERATION	++	+++	++
AORTIC STENOSIS	++	++	+
AORTIC REGURGITATION	++	-	+++
AORTIC ROOT DILATION	+	+++	+++
TUBULAR SEGMENT DILATION	+	++	+++
AORTIC COARCTATION	+	+++	+++

MRI—magnetic resonance imaging. +++ = very positive; ++ = positive; + = fair; - = no.

## Data Availability

Not applicable.

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
