# Peer review of "Unraveling Bicuspid Aortic Valve Enigmas by Multimodality Imaging: Clinical Implications"

_jcm, 2022, doi:10.3390/jcm11020456_

Round 1

Reviewer 1 Report

The aim of this review was to update the role of multimodality imaging techniques in BAV assessment, including recent advances and pending gaps. However there are some issues that should be addressed.

1. TTE is adequate in the vast majority of cases to diagnose a BAV, to ascertain valve morphotype and to assess valve function. Although authors stated that TTE is the first imaging tool, it should be pointed out more clearly that TTE is used for first line imaging for BAV diagnosis, valvular phenotyping and function, measurement of thoracic aorta, exclusion of other aortic malformations, as well as assessment of complications such are infective endocarditis and aortic dissection, thus the readers can have more clearly the role of TTE in everyday clinical practice. Some of it is generally mentioned in the introduction section as the role of imaging.

2. Role of transesophageal echocardiography should be also included in the diagnosis of aortic dissection and infective endocarditis.

4 Add advantages of CMR in comparison to both TTE and CT.

5. Simple BAV imaging evaluation algorithm (together for TTE, CMR and CT) protocol  should be presented and might be helpful for readers. Main anatomical aspects should be pointed out: type and speciphic phenotype of the BAV and the valve function, characteristics of raphe and presence of aortic dilatation

6. Please also add about aortic valve endocarditis, prevalence and how imaging can be useful for diagnosis.  

7. Since this is review of the role of imaging and clinical implications, please add in the "Valvular dysfunction" section for each aortic stenosis and aortic regurgitation the TTE indications for aortic valve replacement.

8. Can CT calcium score might be useful for determining the severity of the valve stenosis? Are there cut off values?

9. In the " 5. Aorta involvement " section please discuss about aortic dissection as the most feared complication of the aorta dilatation. Also add the prevalence of aortic dissection and BAV as well as risk factors.

10. Discuss the role of TAVR in patient with BAV.

11. Repair of the BAV has become an accepted alternative to replacement in patients with BAV regurgitation - add the role of imaging in decision making.

12. Finally, the authors need to clearly highlight what new information this review provides.  

Minor comments:

Please throughout the manuscript correct space between references and text.

Author Response

The aim of this review was to update the role of multimodality imaging techniques in BAV assessment, including recent advances and pending gaps. However there are some issues that should be addressed.

We thank the reviewer for your comments and suggestions.

  1. TTE is adequate in the vast majority of cases to diagnose a BAV, to ascertain valve morphotype and to assess valve function. Although authors stated that TTE is the first imaging tool, it should be pointed out more clearly that TTE is used for first line imaging for BAV diagnosis, valvular phenotyping and function, measurement of thoracic aorta, exclusion of other aortic malformations, as well as assessment of complications such are infective endocarditis and aortic dissection, thus the readers can have more clearly the role of TTE in everyday clinical practice. Some of it is generally mentioned in the introduction section as the role of imaging.

I full agree with the reviewer opinion regarding that TTE is the reference technique in diagnostic and follow-up BAV patients. I thought it was clear throughout the document although the focus of this manuscript is on multimodality imaging. However, owing to aortic root asymmetry, maximum diameter is underestimated > 5mm by TTE compared to CT/MRI diameter in 40% of patients. Based on that, we will recommend in the next ESC Guidelines to perform a baseline CT/MRI in all BAV patients with aortic diameter > 40 mm by TTE. In addition, TTE has a screening role in the diagnosis of associated aortic malformation and aortic dissection being CT the technique of choice. We have added the sentence that you suggested in the abstract, in the valvular dysfunction section and in the new comments on valvular endocarditis.

  1. Role of transesophageal echocardiography should be also included in the diagnosis of aortic dissection and infective endocarditis.

We included the role of TEE in the diagnosis of both complications although in diagnosis of aortic dissection TEE should not been performed except in patient under anaesthesia as it will be stated in the next guidelines. The role of TEE in these complications is not different to the other cases with endocarditis or aortic dissection.

4 Add advantages of CMR in comparison to both TTE and CT.

We have already stated in the section of valvular dysfunction: The evaluation of the AR severity in BAV is challenging by TTE given that eccentric jets are very common. In this setting, CMR may constitute a superior method to quantify both AR severity and LV volumes. 

In addition, CMR also plays a unique role in these two sections: Aortic biomechanical parameters and aortic flow hemodynamic parameter.

Moreover, in the surveillance section we stated: In young patients, if elevated creatinine or contrast allergy CMR would be preferred

  1. Simple BAV imaging evaluation algorithm (together for TTE, CMR and CT) protocol should be presented and might be helpful for readers. Main anatomical aspects should be pointed out: type and specific phenotype of the BAV and the valve function, characteristics of raphe and presence of aortic dilatation.

Thank you for your suggestion. We have included a new figure, figure 6, showing an imaging algorithm for diagnosis and follow-up,

  1. Please also add about aortic valve endocarditis, prevalence and how imaging can be useful for diagnosis.

We have included a new section 9: Complications. Surgical and TAVR indications.

  1. Since this is review of the role of imaging and clinical implications, please add in the "Valvular dysfunction" section for each aortic stenosis and aortic regurgitation the TTE indications for aortic valve replacement.

As we have added a new section with surgical and TAVR indications, we included these indications in that section 9,

  1. Can CT calcium score might be useful for determining the severity of the valve We stenosis? Are there cut off values?

Current valvular guidelines advocate measuring CT calcium score in patients suspected of having severe AS in whom severity cannot be conclusively determined with ETT alone. These recommendations suggest several sex-specific thresholds for men and women, but no specific data have been reported regarding BAV. What it is well known is that for the same age and sex, patients suffering from BAV has more calcificated valves, and that also this increased aortic valve calcium load is significantly correlated with a higher mean transvalvular gradient. This information suggests that more research in this field is needed.

  1. In the " 5. Aorta involvement " section please discuss about aortic dissection as the most feared complication of the aorta dilatation. Also add the prevalence of aortic dissection and BAV as well as risk factors.

We have included information on aorta dissection complication in section 9.

  1. Discuss the role of TAVR in patient with BAV.

We have included TAVR in patients with BAV in section 9.

  1. Repair of the BAV has become an accepted alternative to replacement in patients with BAV regurgitation - add the role of imaging in decision making.

We have discussed indications and role of echocardiography in section 9.

  1. Finally, the authors need to clearly highlight what new information this review provides. 

We have added a final section with the highlights of this review.

Minor comments:

Please throughout the manuscript correct space between references and text.

We have corrected it in the new version.

Reviewer 2 Report

this is a comprehensive review of the role of the various imaging modalities in assessing patients with bicuspid aortic valves. The complementary roles of tte, tee, CT and CMR are discussed in detail.

I believe the manuscript could be made more useful for the audience of this journal if the images shown could be accompanied by  schematic figures emphasizing the relevant information.

Also, a table summarizing the role of the various imaging modalities in ascertaining anatomic and physiologic information wouldd be helpful

There is increasing interest in the role of TAVR in the treatment of patients with bicuspid valves- the authors should discuss the utility of the various imaging modalities in selection of patients for open surgery vs TAVR

Author Response

Reviewer 2.

This is a comprehensive review of the role of the various imaging modalities in assessing patients with bicuspid aortic valves. The complementary roles of tte, tee, CT and CMR are discussed in detail.

Thank you for your comments

I believe the manuscript could be made more useful for the audience of this journal if the images shown could be accompanied by schematic figures emphasizing the relevant information.

 I may understand your suggestion for not cardiologist readers but is impossible for us to prepare these schematic figures. We have submitted many papers with similar figures and they have never required. We have added some comments in the figure legends trying to improve imaging understanding.

Also, a table summarizing the role of the various imaging modalities in ascertaining anatomic and physiologic information would be helpful.

Thank you for your suggestion. We have prepared a table summarizing the role of imaging techniques (table 1)

There is increasing interest in the role of TAVR in the treatment of patients with bicuspid valves- the authors should discuss the utility of the various imaging modalities in selection of patients for open surgery vs TAVR.

This is a new therapeutic option, there are not long-term results but we have included the current reported information.

Round 2

Reviewer 1 Report

Many thanks to the authors who have addressed appropriately all the issues raised by this reviewers. There are still typos in the revised version of the manuscript. Please carefully proofread the manuscript...

Minor comments: 

  1. Row 367, please delete extra "." 
  2. Row 368, please add a reference for a BAV stenosis TAVR Registry. 

Author Response

Thank you so much for your revision.

Changes have been made in the new version of the manuscript.

This manuscript is a resubmission of an earlier submission. The following is a list of the peer review reports and author responses from that submission.